# Hypervirulent *Klebsiella pneumoniae* causing bloodstream infections in Hungary

Fatma A. Mohamed,[1,2] Bálint Timmer,[1,3] Renáta Hargitai,[1] Szilvia Melegh,[1] Réka Meszéna,[1] Tibor Pál,[1] Péter Urbán,[4] Róbert Herczeg,[4] Attila Gyenesei,[4] Ágnes Sonnevend[1,5]

**ABSTRACT**  Hypervirulent *Klebsiella pneumoniae* (hvKP) can cause severe infections even in healthy individuals. Currently, no data are available on the frequency of hvKP-induced bloodstream infections (BSI) in Hungary. Our investigation revealed that of the 157 *K. pneumoniae* isolated from BSI in 2020–2022 at a university hospital in Hungary, three (2%) carried the hypervirulence-associated *rmpA* and *iutAiucABCD* genes. The complete genomes of these three hvKP isolates were sequenced. They were unrelated and belonged to ST5, ST86, and ST6771, a single-locus variant of ST893, i.e., to internationally known hvKP clones. In the *K. pneumoniae* ST86 and ST6771 isolates, the *rmpA/A2*, aerobactin, and salmochelin siderophore genes were located on virulence plasmids highly similar to those of *K. pneumoniae* ST23 and ST86 isolated in Asia, while the *K. pneumoniae* ST5 isolate harboured *rmpA*, *iroBCDN*, and yersiniabactin locus on a chromosomally integrated ICE*Kp1* element. Comparison of the core genome MLST of the three Hungarian hvKP isolates to genomes belonging to the same ST/CC deposited in the Bigsdb database of the Pasteur Institute revealed that, although no direct epidemiological link could be established, KP48326 *K. pneumoniae* ST86, isolated in Pécs, clustered with a Greek isolate (ID-48733). The emergence of *K. pneumoniae* belonging to known hypervirulent clones in Hungary, albeit sporadic, is alarming and underscores the importance of continued whole-genome-based epidemiological surveillance.

**IMPORTANCE** This study represents the first investigation of the prevalence of hypervirulent *K. pneumoniae* (hvKP) in bloodstream infections in Hungary, conducted at the University Hospital of Pécs. Our findings emphasize the need to accurately identify hvKP strains, integrating both phenotypic and genotypic screening. Whole genome sequencing revealed genetic diversity among the Hungarian hvKP isolates, confirming the emergence of globally disseminating hvKP clones—ST86, CC893, and ST5—in Hungary. The localization of hypervirulence-related genes on mobile genetic elements, e.g., on virulence plasmids or on ICE*Kp1* similar to those found in hvKP isolates from different continents, underscores the significant role of horizontal gene transfer in the spread of hvKP. Overall, the study enhances our understanding of hvKP epidemiology and underscores the importance of continued molecular surveillance and control measures to mitigate the threat of hvKP infections in Hungary.

**KEYWORDS**  *Klebsiella pneumoniae*, hypervirulent clones, bloodstream infections

*K*lebsiella pneumoniae is a Gram-negative bacterium that causes community-acquired and healthcare-associated illnesses, such as pneumonia, urinary tract, and bloodstream infections particularly in immunocompromised individuals (1, 2). The virulence of a given *K. pneumoniae* strain causing an infection considerably impacts morbidity and mortality (3). The type of capsule, lipopolysaccharide (LPS), siderophores,

**Peer Reviewer** Maria Dolores Alcántar-Curiel, Facultad de Medicina, Universidad Nacional Autónoma de México, Mexico City, Mexico

Address correspondence to Ágnes Sonnevend, pal.agnes@pte.hu.

The authors declare no conflict of interest.

See the funding table on p. 9.

urease, and fimbriae are factors associated with virulence promoting adherence, immune evasion, and nutrient scavenging (3).

Unlike the mainly opportunistic "classical" *K. pneumoniae* (cKP) strains, a novel variant, termed "hypervirulent" (hvKP) infecting previously healthy individuals, has emerged over the last three decades, particularly in the Asian Pacific Rim (4, 5). Even though the distinction between cKP and hvKP is not always obvious, they frequently differ in their demographic and clinical distribution (3). HvKP typically causes community-acquired infections, such as primary hepatic abscess, necrotizing fasciitis, endophthalmitis, and meningitis in healthy people of any age. The mortality of hvKP bacteraemia is significantly higher compared to bacteraemia caused by non-hvKP (6). Cases of community-acquired pneumonia with bacteremia and necrotizing fasciitis have also presented with high mortality rates of 55% and 47%, respectively. Furthermore, hvKP infections often cause permanent loss of vision or neurologic sequelae (7). The hvKP strains are often hypermucoviscous due to overproduction of capsular polysaccharides and produce several different siderophores (8). Nevertheless, although the terms hypermucoviscous and hypervirulent KP have frequently been used as synonyms, mucoviscosity, as detected by the string test (9), *per se* cannot safely predict hypervirulence (8). Instead, detection of genes, such as *iucA*, *peg-344*, and *rmpA/A2*, associated with hvKP, is a more precise way of identifying this pathotype (10–12).

Lately, hvKP infections have increasingly been recorded in Europe (13), Africa (14), Australia (15), South America (16), and North America (17). The true prevalence of hvKP is difficult to assess as studies defined hvKP based on various combinations of markers. Nevertheless, when considering only genetic marker-defined hvKP prevalence among bloodstream isolates, the highest prevalence was observed in China, where it reached 73.9% in certain regions (18), and in South and Southeast Asian countries, with a prevalence around 20% (19), whereas in North America and Europe, it was found to be invariably <10% (3.8% in Chicago, USA [20], 8.2% in Canada [21], 3.2% in Barcelona, Spain [22], 1.3% in Italy [23], and 3.3% in Croatia [24]).

Except for an innate resistance to ampicillin, hvKP isolates originally had been susceptible to most routinely used antibiotics (10). However, in recent years, multidrug-resistant (MDR) hvKP isolates have emerged, particularly in China (25, 26). The convergence of resistance and virulence poses a formidable challenge to treatment, especially when hvKP is producing a carbapenemase enzyme. Carbapenem-resistant hvKP belonging to sequence type ST23 also occurred in several European countries, including one case being reported in 2023 from Hungary (27).

Nevertheless, the prevalence of hvKP among bloodstream isolates has not yet been studied systematically in Hungary. Therefore, the aim of our investigation was to assess the presence and characteristics of hvKP and to determine its proportion among bloodstream isolates in our university hospital, where *Klebsiella pneumoniae* was a significant pathogen, isolated from 8.7% of true-positive blood cultures during the study period.

## RESULTS

### Identifying *Klebsiella pneumoniae* with hypervirulent features

Altogether, 14 strains exhibited the hypermucoviscous phenotype; nevertheless, twelve of them did not possess the *rmpA/A2* and *iucA* genes, which were present in two string test positive isolates (KP48326 and KP873) and in another non-hypermucoviscous isolate (KP7389). As the hyperviscous character alone does not define hvKP (11), only the three PCR-positive strains for rmpA/A2 and iucA were selected for further studies. They represented 2% of all *K. pneumoniae* bloodstream isolates encountered.

The details of the patients from which these isolates were recovered, and the characteristics of the strains are presented in Table 1. Infections caused by hvKP strains had been present at admission in two patients (strains KP48326 and KP873), both of whom were admitted for community-acquired pneumonia and sepsis. Bloodstream infection symptoms in the third patient appeared on the fourth day after admission;

**TABLE 1** Characteristics of patients with hypervirulent *Klebsiella pneumoniae* bloodstream isolates identified[a]

| Isolate | Age (years) | Sex | Ward | Admission | Specimen collection | Site of original infection | Clinical signs of sepsis | Underlying conditions | AB therapy | Outcome |
|---|---|---|---|---|---|---|---|---|---|---|
| KP48326 | 45 | F | Pulmonology | 10/Dec/2020 | 10/Dec/2021 | CAP | Yes | HT, DM type 2, benzodiazepine intoxication | CRO, AZI | Discharged |
| KP873 | 80 | F | Emergency Dept. | 7/Jan/2021 | 7/Jan/2022 | CAP | Yes | esophageal achalasia | CRO | Deceased due to hospital-acquired SARS-CoV-2 infection |
| KP7389 | 72 | F | Cardiac Surgery | 16/Feb/2021 | 19/Feb/2021 | BSI | Yes | CAD, CABG surgery with ECCMO | AMC | Discharged |

[a]CAP: community-acquired pneumonia; BSI: bloodstream infection; HT: hypertonia; DM: diabetes mellitus; CAD: coronary artery disease; CABG: coronary artery bypass graft; ECCMO: extracorporeal membrane oxygenation; CRO: ceftriaxone,;AZI: azithromycin; AMC: amoxicillin-clavulanic acid.

hence, this case was regarded as hospital-acquired. It should be noted, however, that based on the data available, the possibility that she had already been carrying the hvKP isolate upon admission cannot be excluded. Although all three patients had sepsis, none developed septic shock or metastatic infections, and the sepsis is manageable with the empiric antibiotic therapy as shown in Table 1.

## Characterization of the hypervirulent *K. pneumoniae* isolates

Except for ampicillin, all three hvKP isolates were susceptible to all tested antibiotics and did not carry any acquired antimicrobial resistance genes. The three isolates belonged to distinct sequence types and possessed different capsular loci, as determined from their whole-genome sequence data (Table 2). *K. pneumoniae* isolates KP48326 and KP7389 carried the hypervirulence-associated genes (*rmpA*, *rmpA2*, and the salmochelin and aerobactin loci) on plasmids. In *K. pneumoniae* KP873, the *rmpA* gene was located on a chromosomally integrated ICE*Kp1* element along with the yersiniabactin and salmochelin siderophore genes (Fig. 1).

Two of the three hvKP isolates identified in the current study belonged to *K. pneumoniae* sequence types described as hypervirulent in Europe, i.e., ST86 and ST5 (23) (Table 2). The third isolate belonged to ST6771, a single-locus variant of ST893, which was reported as hvKP from Iran (28). A comparison of the cgMLST profiles of the Hungarian isolates with genomes of the same ST/CC deposited in the Pasteur Institute's Bigsdb database revealed that only isolate KP48326, a *K. pneumoniae* ST86 strain, clustered with an international isolate (id-48733 from Greece) with a 13-allele difference, while the other two Hungarian hvKP isolates did not exhibit close relationships with any international strains (Fig. 2A through C). Further assessment of the genomic relatedness

**TABLE 2** Characteristics of hvKP bloodstream isolates based on their whole genome sequences

| Laboratory number | MLST | Predicted O serotype | Capsular locus | Chromosome / plasmid | GenBank acc. no. | Size (bp) | Inc type | Virulence genes | Antibiotic resistance genes |
|---|---|---|---|---|---|---|---|---|---|
| KP48326 | ST86 | O1 | KL2 | Chromosome | CP167190 | 5187926 | NA | None | bla$_{SHV-1}$ |
| | | | | pKP48326-1 | CP167191 | 226735 | IncHI1B, repB_KLEB | *rmpA, rmpA2,* aerobactin, salmochelin loci | None |
| KP873 | ST5 | O1 | KL39 | Chromosome | CP163248 | 5320900 | NA | rmpA, yersiniabactin, salmochelin loci | bla$_{SHV-62}$, *fosA, oqxAB* |
| | | | | pKP873-1 | CP163249 | 191144 | IncFIB | Aerobactin locus | None |
| | | | | pKP873-2 | CP163250 | 41366 | NT | None | None |
| KP7389 | ST6771 (CC893) | O3 | KL20 | Chromosome | CP167192 | 5244428 | NA | Yersiniabactin locus | bla$_{SHV-26}$, *fosA6, oqxAB* |
| | | | | pKP7389-1 | CP167193 | 212430 | repB_KLEB | *rmpA, rmpA2,* aerobactin, salmochelin loci | None |
| | | | | pKP7389-2 | CP167194 | 111189 | IncFIB (pKPHS1) | None | None |

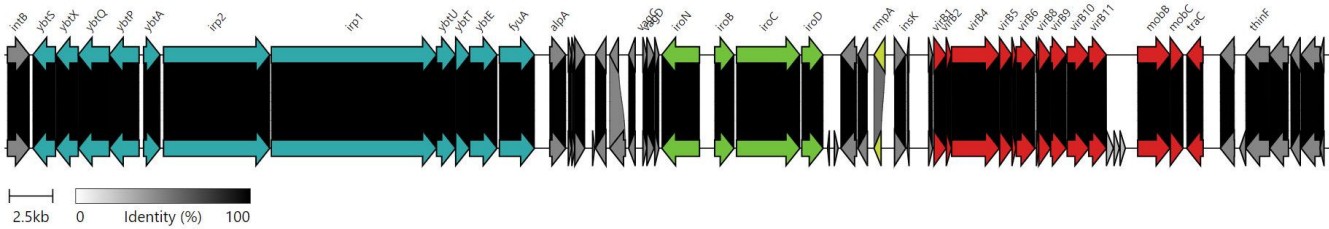

**FIG 1** Comparison of the ICE*Kp1* sequence and the corresponding region in the *K. pneumoniae* KP873 chromosome. The upper sequence is GenBank accession no. KY454627. The lower is the corresponding region of *K. pneumoniae* KP873 chromosome. Similarity is marked with gray scale. Red color marks genes responsible for mobilization, turquoise marks yersiniabactin locus, green salmochelin locus, and yellow the *rmpA* gene.

of KP48326 and id-48733 by Mash (d-distance), FastANI, and core-genome single-nucleotide polymorphism (SNP) analysis consistently indicated a high degree of genomic similarity, with a low d-distance (0.000336872), high FastANI identity of 99.9977%, and only 43 SNP differences, collectively supporting that the genomes are indeed highly similar.

## Comparison of the hypervirulence plasmids

The two plasmids from strains KP48326 and KP7389, i.e., pKP48326-1 and pKP7389-1, carrying the hypervirulence-associated genes (i.e., *rmpA*, *rmpA2*, *peg-344*, *iucA*, and *iutABCD*) belonged to the PTU-E21 plasmid taxonomic unit. Upon examining the $ANI_{L50}$ network of the entire prokaryotic plasmidome (29), we found that only eight PTU-E21 plasmids were present in a distinct plasmid taxonomic unit within the broader panplasmidome network.

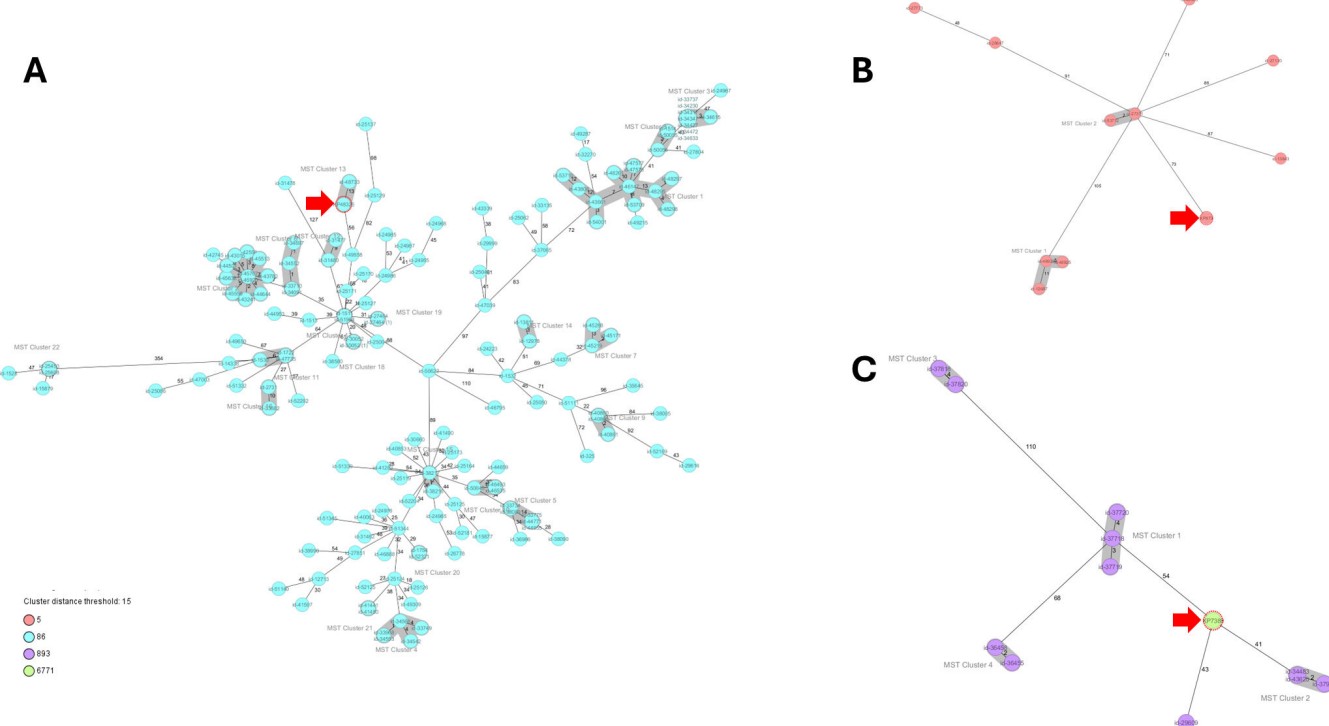

**FIG 2** Minimum spanning trees based on cgMLST of *K. pneumoniae* ST5, ST86, and CC893 strains. Isolates from this study are marked with red arrows. (A) MST of *K. pneumoniae* ST86. (B) MST of *K. pneumoniae* ST5. (C) MST of *K. pneumoniae* CC893. Distances are based on columns from *K. pneumoniae sensu lato* cgMLST (2358) of the Ridom SeqSphere+. Color of nodes corresponds to sequence types (STs) as shown in the lower left corner of the figure. Clusters were defined as isolates exhibiting 15 or fewer allelic variations.

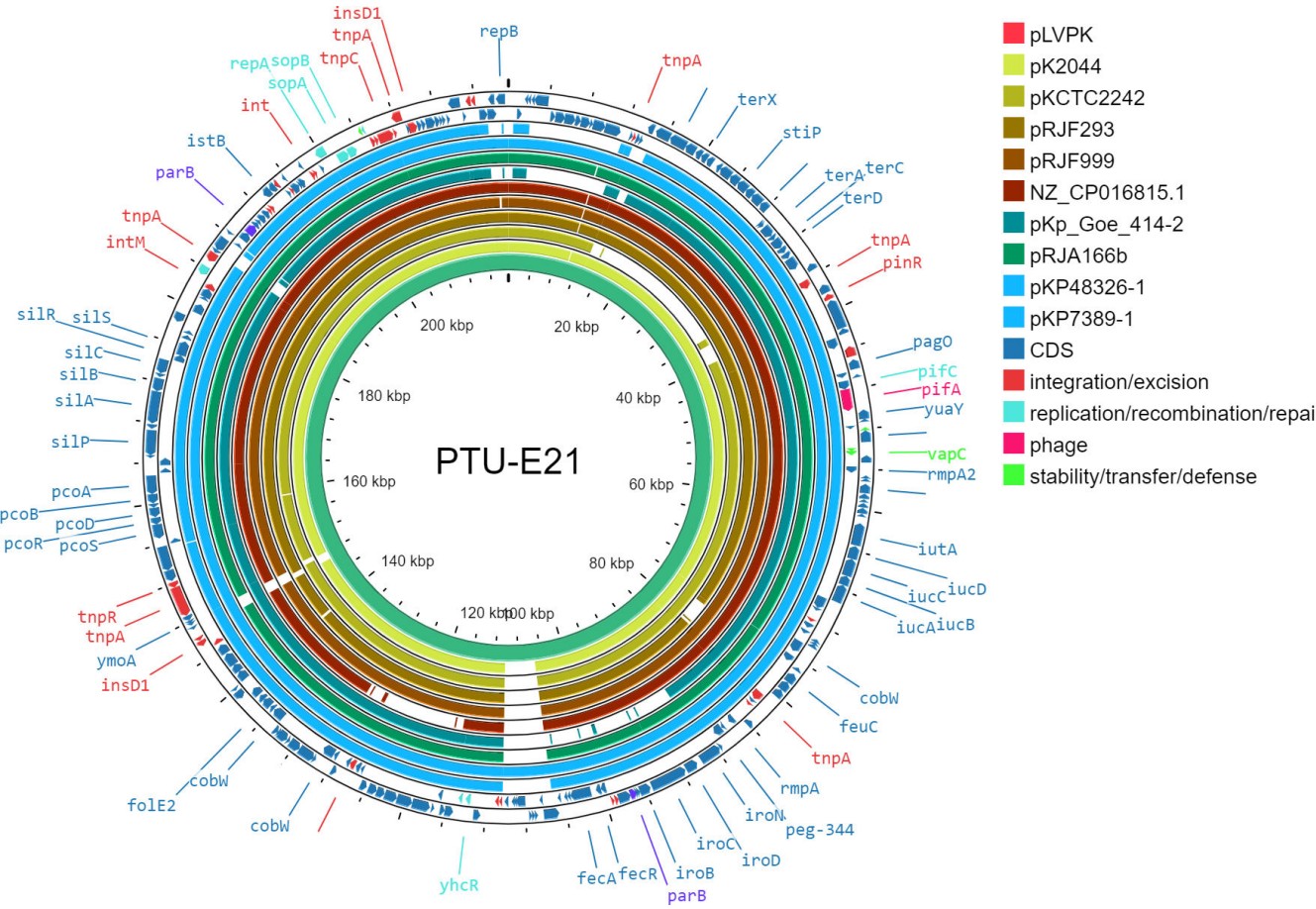

**FIG 3** Circular comparison of the ten PTU-E21 plasmids by Proksee. Circles of various colors represent individual hypervirulence plasmids of the PTU-E21. Coding sequences (CDS) are marked by arrows. Gene names are written with the color code specified on the figure.

Comparison of the ten PTU-E21 plasmids by Proksee is shown in Fig. 3. Plasmids pKP48326-1 and pKP7389-1 and the eight PTU-E21 plasmids in the network (listed in Table 3) had Mash distances below 0.01 (indicating >99.9% sequence similarity) (Fig. S1). Although untypeable by pubMLST, the closest match to these plasmids was the IncFIA 27 allele variant. The plasmids possessed intact replication and partitioning systems, but

**TABLE 3** Characteristics of the plasmids belonging to PTU-E21

| GenBank accession number | Plasmid | Size (bp) | pMLST | Host strain | Host species | Sequence type | Country | Year of isolation | Isolation source |
|---|---|---|---|---|---|---|---|---|---|
| NC_005249.1 | pLVPK | 219385 | ~FIA:27 | CG43 | *K. pneumoniae* | 86 | Taiwan | 2013 | Liver abscess |
| NC_006625.1 | pK2044 | 224152 | ~FIA:27 | NTUH-K2044 | *K. pneumoniae* | 23 | Taiwan | 2005 | Blood |
| NC_017541.1 | pKCTC2242 | 202852 | ~FIA:27 | KCTC 2242 | *K. pneumoniae* | 375 | South Korea | 2011 | Unknown |
| NZ_CP014009.1 | pRJF293 | 224263 | ~FIA:27 | RJF293 | *K. pneumoniae* | 374 | China | 2016 | Blood |
| NZ_CP014011.1 | pRJF999 | 228907 | ~FIA:27 | RJF999 | *K. pneumoniae* | 23 | China | 2015 | Blood |
| NZ_CP016815.1 | unnamed | 212770 | ~FIA:27 | ED23 | *K. pneumoniae* | 23 | Taiwan | 2006 | Blood |
| NZ_CP018338.1 | pKp_Goe_414-2 | 202175 | ~FIA:27 | Kp_Goe_154414 | *K. pneumoniae* | 23 | Germany | 2014 | Wound swab |
| NZ_CP019049.1 | pRJA166b | 228613 | ~FIA:27 | RJA166 | *K. pneumoniae* | 23 | China | 2015 | Sputum |
| CP167193 | pKP7389-1 | 212430 | ~FIA:27 | KP7389 | *K. pneumoniae* | 6771 | Hungary | 2021 | Blood |
| CP167191 | pKP48326-1 | 226735 | ~FIA:27 | KP48326 | *K. pneumoniae* | 86 | Hungary | 2020 | Blood |

they lacked the genes necessary for conjugation (Fig. 3), as characterized by Mobtyper, which identified them as non-mobilizable due to the absence of MOB and MPF regions.

## DISCUSSION

Globally, infections caused by hypervirulent *Klebsiella pneumoniae* (hvKP) isolates have increased (3, 10, 30). Here, we report that in our clinical settings in Hungary, the prevalence of hvKP (defined by the possession of capsular overproduction [*rmpA/A2*] and aerobactin [*iutAiucABCD*] genes) is 2% among *K. pneumoniae* bloodstream isolates. This figure is comparable to multi-center data from Italy (1.3%) (23), but lower than what was observed in Croatia (3.3%) (24), a country bordering Southwestern Hungary—the region where our university hospital is located—and markedly lower than what was reported from China, South Korea, Taiwan and South Africa (22.8%, 42.2%, 46% and 52%, respectively) (31–33). It is of note, though, that to the best of our knowledge, other countries in Central Europe, *e.g.* Austria, Slovakia, Romania, and Serbia have not reported the presence of hvKP yet. The epidemiological difference could partly be explained by increased colonization and transmission, and possible, but not proven, genetic predisposition of people living in or originating from Southeast Asian countries (34).

The hvKP isolates identified in this study belonged to established hypervirulent clones, i.e. ST5, ST86, and ST6771, the latter being a single-locus variant of ST893. Notably, we did not identify any hvKP ST23, the most common hvKP clone in Southeast Asia, and also found in other European countries (Germany, Italy, Poland, Switzerland, France, Ireland, Lithuania, Latvia and the Netherlands) (27, 35). However, as our collection was restricted to bloodstream isolates, the possibility of hvKP ST23 being present in liver abscess, cerebrospinal fluid, necrotizing fasciitis, or endophthalmitis samples cannot be excluded. Of the hvKP clones identified in this study, ST86 is the most widely recognized, having been reported in several countries, including Croatia (24), France (36), Spain (22), Japan (37), China (38), and the USA (39). Although cgMLST and genomic analysis identified a Greek hvKP ST86 closely related to KP48326, our patient's history did not reveal any connection to Greece. Publication on hvKP ST86 in Europe is available on sporadic cases detected in France, Spain, Italy, and Croatia (22–24, 40). HvKP ST893 is a predominant clone in Iran, and although the Pasteur BigSdb database contains hvKP genomes from Belgium and Norway (Table S1), to the best of our knowledge, no publication describes such isolates in Europe (28). Both ST86 and ST893 clones usually possess the hypervirulence determinant on plasmids, similar to our isolates (KP48326 and KP7389) (37, 38). In our ST86 and ST6771 (CC893) hvKP isolates, the hypervirulence plasmids belonged to the plasmid taxonomic unit PTU-E21. These plasmids showed high homology based on their Mash distances, which is suggestive of a recent common ancestor. Furthermore, all PTU-E21 plasmids (Table 3) showed the closest match to the IncFIA 27 allele in pubMLST, possibly evolving from an IncF plasmid lineage. These >200 kb large plasmids harbor several transposons and, in some instances, encode putative genes that show homology to chromosomal genes, inferred from protein homology data. This suggests that recombination events may have contributed to the evolution of their ancestral plasmid. While these plasmids possess intact replication and partitioning systems, they lack the genes necessary for conjugation, as revealed by Mobtyper. The lack of conjugation machinery raises the possibility that alternative mechanisms may be responsible for their dissemination, particularly given that these plasmids were found in strains not clonally related. This combination of high sequence similarity, broad geographic distribution, and absence of conjugation mechanisms suggests a complex interaction of factors driving the distribution and persistence of these plasmids in bacterial populations.

The third hvKP isolate encountered in our study was a *K. pneumoniae* ST5 strain, carrying the hypervirulence-determining genes on a chromosomally located integrative conjugative element, ICE*Kp1*. Hypervirulent *K. pneumoniae* of this sequence type is seldom reported, mainly from Italy (23, 41), but a few genomes of hvKP isolates from

the United Kingdom, Ireland, Spain, and Slovenia are deposited in the Pasteur BigSdb database (Table S1). A member of this clone with chromosomally encoded hypervirulence determinants (i.e., an isolate similar to the one described here) was highly virulent in *in vivo* models (23).

In general, our findings are in line with previous observations that community acquisition and antibiotic susceptibility characterize the majority of hvKP strains (31). Nevertheless, interestingly, in our setting, one isolate—*K. pneumoniae* KP7389, an ST6771 (CC893) strain—could be considered hospital-acquired.

Furthermore, contrary to the findings from Southeast Asia (42), all three patients in our clinical center with hvKP bloodstream infections were of advanced age and had multiple underlying comorbidities (Table 1). Despite this, hvKP pulmonary infections and/or sepsis were successfully managed with the empirical antibiotic therapy, and septic shock or metastatic infections did not occur in them. The single patient who succumbed did so due to severe SARS-CoV-2 pulmonary superinfection, likely acquired during hospitalization, as she tested negative for SARS-CoV-2 at admission and became positive only on the 25th day of her hospital stay.

The European Centre for Disease Prevention and Control (ECDC) issued a warning in 2021 (27) on the emergence of carbapenem-resistant hypervirulent *K. pneumoniae* in Europe. Such isolates have been reported from Western Europe (Spain, the United Kingdom, Italy, France, and Switzerland) and Eastern Europe (Poland, Latvia, Lithuania, Croatia, and Hungary) (42, 43), the USA (39), and areas where hvKP was already endemic, e.g., China (25, 30, 38). In the collection of bloodstream *K. pneumoniae* isolates from the university hospital, no carbapenem-resistant hvKP isolates were identified, and the strains exhibited a broad susceptibility spectrum (data not shown). As other parts of the country experience a higher rate of carbapenem-resistant *K. pneumoniae* clinical isolates, the likelihood of recovering resistant hvKP is higher elsewhere in Hungary. The emergence of such isolates has recently been reported in a non-abstracted, Hungarian-language conference presentation (Á. Tóth, Kristóf K., Hanczvikkel A., Buzgó L., Ungvári E., Tóth K., Göbhardter D., and Damjanova I. *presented at the Scientific Meeting of the Microbiology Section of* the Hungarian Society of Infectious Diseases and Clinical Microbiology, Budapest, Hungary, 7 March 2024). While a limitation of our study is that it was conducted in a single center, it nonetheless confirmed the presence of international hvKP clones; hence, it emphasizes the necessity of continuous, ongoing monitoring for the presence of hypervirulent *K. pneumoniae*. Such surveillance, based on testing of all *K. pneumoniae* from bloodstream infections and other likely infectious sites, e.g., liver abscess, endophthalmitis, and cerebrospinal fluid, using genetic markers, would be beneficial to assess the true burden of hvKP in Hungary and other countries with low prevalence.

## MATERIALS AND METHODS

### Strain collection

Between January 2020 and August 2022, 157 *K. pneumoniae* isolates—the first blood culture isolate from individual patients treated at the 1,459-bed tertiary care hospital of the Clinical Center of the University of Pécs—were collected. Strains were identified by matrix-assisted laser desorption ionization time-of-flight mass spectrometry (MALDI-TOF MS) (Bruker Daltonics, Bremen, Germany) and stored at −80°C in Tryptic Soy Broth (TSB, MAST, UK) containing 20% glycerol until further investigations.

### Screening for hypervirulent *K. pneumoniae*

The string test was used to test the hypermucoviscous phenotype. Briefly, isolates grown overnight at 37°C on a sheep blood agar plate were tested for the development of a ≥5 mm viscous filament if pulled upward by an inoculation loop (17). Simultaneously, the

presence of rmpA, rmpA2, and iucA genes was tested by PCR as described in (17, 44, 45), respectively.

## Antimicrobial susceptibility testing

The susceptibility of strains identified as hvKP to ampicillin, ampicillin-sulbactam, amoxicillin-clavulanate, piperacillin-tazobactam, cefuroxime, cefotaxime, ceftazidime, ceftriaxone, cefepime, ceftazidime-avibactam and ceftolozane-tazobactam, ertapenem, imipenem, meropenem, ciprofloxacin, levofloxacin, gentamicin, tobramycin, amikacin, trimethoprim-sulfamethoxazole, and cefiderocol was tested by Kirby–Bauer disk diffusion method on Mueller–Hinton agar (BioRad, Marnes-la-Coquette, France), according to the EUCAST guidelines (46).

## Complete genome sequencing and analysis

The sequences of the complete genomes of strains identified as hvKP were determined by 150 bp paired-end sequencing on Illumina NovaSeq platform and long-read sequencing on Oxford Nanopore MinIon or on PacBio platforms. Hybrid assembly of short and long reads was performed using Unicycler v.0.5.0 (47). The quality control of the reads and assemblies is shown in Tables S2 to S7.

The capsular locus and the multi-locus sequence type of the isolates were deduced from their genome sequence on the PathogenWatch website (23). Whole genomes of *K. pneumoniae* belonging to the same sequence types as the hvKP isolates identified in this study were downloaded from the Bigsdb database of the Pasteur Institute (accessed on the 03/07/2023), and the cgMLST of the Hungarian isolates was compared to those with similar STs deposited in the database using the Ridom SeqSphere software (the details of these genomes are listed in Table S1). Genetic distance of isolates from this study and from the Bigsdb database clustering with ≤15 allelic differences was analyzed using three complementary approaches: Mash (v2.1) for estimating genetic distance (d-distance) (48), FastANI (v1.34) for pairwise nucleotide similarity (https://github.com/ParBLiSS/FastANI/releases/tag/v1.34), and Snippy (v4.0.2) for single-nucleotide polymorphism (SNP) (https://github.com/tseemann/snippy).

Plasmid replicon types, resistance, and virulence genes were identified using ABRicate (https://github.com/tseemann/abricate) with PlasmidFinder (49), applying a minimum identity threshold of 95%, as well as CARD and VFDB databases (50, 51), respectively. MobileOG-db (52) was used for mobile genetic element prediction as an integrated tool in Proksee (53). The plasmid taxonomic unit (PTU) of the identified hypervirulence plasmids was determined using COPLA (29). Plasmids found to belong to the same PTU in the $ANI_{L50}$ network of the entire prokaryotic plasmidome (54) were compared to the plasmids identified in our study using the MinHash algorithm via Mash (48), followed by the creation of a Mash distance matrix. The matrix was visualized as a heatmap using Matplotlib. Gene annotation was performed using Bakta v1.9.1 (55). Subsequently, we performed nucleotide BLAST comparisons involving our plasmids and those belonging to the same PTU using the Proksee pipeline. All plasmid sequences are obtained from the NCBI database as shown in Table 3. The circular comparison was visualized by Proksee (52). *In silico* mobility prediction, including the screening of MOB and MPF types, was conducted using Mobtyper (56).

## ACKNOWLEDGMENTS

This work was supported by the University of Pécs Medical School's Kispál Gyula grant no. 300852 to A.S. F.A.M. is supported by the Stipendium Hungaricum Scholarship. The funders had no role in the study design, data collection and interpretation, or decision to submit the work for publication.

## AUTHOR AFFILIATIONS

[1]Department of Medical Microbiology and Immunology, University of Pécs Medical School, Pécs, Hungary

[2]Department of Microbiology and Immunology, Faculty of Pharmacy, Zagazig University, Zagazig, Egypt

[3]Department of Metagenomics, University of Debrecen, Debrecen, Hungary

[4]Hungarian Centre of Genomics and Bioinformatics, Szentágothai Research Center, University of Pécs, Pécs, Hungary

[5]Molecular Medicine Research Group, Szentágothai Research Center, University of Pécs, Pécs, Hungary

## AUTHOR ORCIDs

Ágnes Sonnevend http://orcid.org/0000-0002-7065-1736

## FUNDING

| Funder | Grant(s) | Author(s) |
| --- | --- | --- |
| Stipendium Hungaricum | | Fatma A. Mohamed |
| University of Pécs Medical School Kispál Gyula Grant | 300852 | Ágnes Sonnevend |

## AUTHOR CONTRIBUTIONS

Fatma A. Mohamed, Data curation, Formal analysis, Investigation, Writing – original draft | Bálint Timmer, Data curation, Formal analysis, Investigation, Methodology, Software, Visualization, Writing – review and editing | Renáta Hargitai, Investigation, Methodology, Writing – review and editing | Szilvia Melegh, Formal analysis, Investigation, Writing – review and editing | Réka Meszéna, Investigation, Methodology, Project administration, Writing – review and editing | Tibor Pál, Formal analysis, Writing – review and editing | Péter Urbán, Investigation, Validation, Writing – review and editing | Róbert Herczeg, Data curation, Formal analysis, Writing – review and editing | Attila Gyenesei, Project administration, Supervision, Validation, Writing – review and editing | Ágnes Sonnevend, Conceptualization, Formal analysis, Funding acquisition, Project administration, Supervision, Writing – review and editing

## DATA AVAILABILITY

The complete genomes were uploaded to GenBank under the accession numbers CP163248-CP163250 (KP873), CP167190-CP167191 (KP48326), and CP167192-CP167194 (KP7389).

## ADDITIONAL FILES

The following material is available online.

### Supplemental Material

**Supplemental figure and tables (Spectrum00031-25-s0001.pdf).** Fig. S1 and Tables S1 to S7.

### Open Peer Review

**PEER REVIEW HISTORY (review-history.pdf).** An accounting of the reviewer comments and feedback.

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
