## [Reviewer comments · Microbiology Spectrum]

Microbiology Spectrum

Hypervirulent *Klebsiella pneumoniae* causing bloodstream infections in Hungary

Fatma Mohamed, Bálint Timmer, Renáta Hargitai, Szilvia Melegh, Réka Meszéna, Tibor Pál, Péter Urbán, Róbert Herczeg, Attila Gyenesei, and Agnes Sonnevend

Corresponding Author(s): Agnes Sonnevend, Pecsí Tudományegyetem Általános Orvostudományi Kar

Review Timeline:

Submission Date:	January 6, 2025
Editorial Decision:	March 31, 2025
Revision Received:	June 1, 2025
Accepted:	June 17, 2025

Editor: Ana Cabrera

Reviewer(s): Disclosure of reviewer identity is with reference to reviewer comments included in decision letter(s). The following individuals involved in review of your submission have agreed to reveal their identity: Maria Dolores Alcántar-Curiel (Reviewer #2)

Transaction Report:

DOI: <https://doi.org/10.1128/spectrum.00031-25>

Re: Spectrum00031-25 (Hypervirulent *Klebsiella pneumoniae* causing bloodstream infections in Hungary)

Dear Dr. Agnes Maria Sonnevend:

Thank you for the privilege of reviewing your work. Below you will find my comments, instructions from the Spectrum editorial office, and the reviewer comments.

Revision Guidelines

Sincerely,
Ana Cabrera
Editor
Microbiology Spectrum

Reviewer #1 (Comments for the Author):

This study examines the prevalence of hypervirulent *Klebsiella pneumoniae* (hvKP) bloodstream infections in Hungary, identifying three hvKP isolates among 157 BSI *K. pneumoniae* strains. Whole-genome sequencing revealed that the strains belonged to globally recognized hvKP lineages (ST5, ST86, and ST6771) and carried hypervirulence-associated genes on plasmids or chromosomally integrated elements. The findings highlight the sporadic but concerning emergence of hvKP clones in Hungary and stress the importance of ongoing genomic surveillance to track the spread of these pathogens. Despite high virulence, all isolates remained antibiotic-susceptible, distinguishing them from multidrug-resistant hvKP strains seen elsewhere.

Generally, I do not have many concerns or comments. I only recommend that the authors inform readers what they should interpret from these findings.

Major Concerns

1. I would encourage the authors to include their takeaway from their findings and what the implications are. hvKP numbers were low in Hungary compared to China, South Korea, Taiwan, and South Africa - any ideas as to why? Same comment but for the similarity to Italy. What should epidemiologists do as a result of these findings? Should we be screening for hvKP genes? Since these infections are generally community acquired, what approach should be taken to study them?

Minor Concerns

1. A comment about limiting study isolates to BSI would be beneficial for context.
2. Was the hospital environment near these patients sampled to determine if the hvKP had colonized any surfaces or wastewater drains?

Reviewer #2 (Comments for the Author):

Abstract

Line 20: "There is no data" change to "no data are available".

Line 22: Add a comma after "Hungary"

Lines 23-24: "three hvKP isolate was determined" correct to "three hvKP isolates were sequenced".

Line 24: Add a comma after "ST86".

Line 26: Add a comma after "isolates".

Line 32 "although epidemiological link could not be established..." modify to "although no direct epidemiological link could be established..."

Line 35: " Highlights the importance of continued whole genome based epidemiological surveillance" modify to " Underscores the importance of continued whole-genome-based epidemiological surveillance".

IMPORTANCE

Line 43: "The localisation of hypervirulence related genes" correct to "The localization of hypervirulence-related genes".

Line 42: Correct "hvKp" to "hvKP".

Line 43-44: Rewrite the following text: "Highly similar to virulence plasmids or chromosomally integrated ICEKp1 of *K. pneumoniae* ST23 and ST86 isolated in other continents".

Line 46 I suggest changing "emphasizes the additional role of horizontal gene transfer" to "Underscores the significant role of horizontal gene transfer in the spread of hvKP."

INTRODUCTION

Line 50: "a Gram-negative bacterium causes" change to "is a Gram-negative bacterium that causes....".

Line 51: Add a comma after "urinary tract and infections".

Line 52: "the virulence of the actual *K. pneumoniae* strain" modify to "the virulence of a given *K. pneumoniae* strain".

Line 60 "HvKP typically causes community acquired infections" change to "HvKP typically causes community-acquired infections".

Line 61: Add a comma after "endophthalmitis".

Lines 69-70: "However, recently the emergence of multi-drug resistant hvKP isolates has been reported mostly from China" modify to "However, in recent years, multidrug-resistant hvKP isolates have emerged, particularly in China".

Line 62: "of capsular polysaccharide" correct to " of capsular polysaccharides".

Although the study focuses on molecular epidemiology, I suggest, it is necessary to expand on the clinical relevance of hvKP by providing information on prevalence rates, mortality, and morbidity of hvKP infections, as well as their therapeutic challenges, particularly in MDR strains.

Information on the prevalence of *K. pneumoniae* infections in Hungarian hospitals should be added, along with any prior reports of hvKP in the region. These data and their comparison with other regions would provide stronger justification for the study in Hungary.

RESULTS

Line 80: Clarify that the hypermucoviscous phenotype alone is not sufficient to define hvKP, as there are hypermucoviscous strains that are not hypervirulent and vice versa.

Line 84: Delete "as well as" and replace it with "were recovered and".

Line 85: Correct "Symptoms of infections from which the hvKP strains was isolated" to "Infections caused by hvKP strains"

Line 87: Change community acquired" to "community-acquired".

Lines 87-88: To improve readability, I suggest changing "Signs of a bloodstream infection of the third patient emerged and the sample was collected on the fourth day after admission" to "Bloodstream infection symptoms in the third patient appeared on the fourth day after admission".

Line 89: Change "hospital acquired" to "hospital-acquired".

Lines 93-94: For clarity, I suggest changing: "Apart from being resistant to ampicillin, all three hvKP isolates remained susceptible to all antibiotics tested and none of them carried any acquired antibiotic resistance genes" to "Except for ampicillin, all three hvKP isolates were susceptible to all tested antibiotics and did not carry any acquired antimicrobial resistance genes".

Line 97: Change "hypervirulence associated" to "hypervirulence-associated".

Line 115: Change "Comparison" to "A comparison".

Since the three hvKP strains were phenotypically resistant only to ampicillin, no convergence of hypervirulence with multidrug resistance was observed. It is necessary to provide patient data to determine whether they developed sepsis or septic shock, if they had complications such as metastatic abscesses, meningitis, or endophthalmitis, and their clinical outcomes.

DISCUSSION

Line 124: Change "have been increasing" to "have increased".

Line 126: Change "hvKP identified" to "hvKP is identified".

Line 137: Change "similarly" to "similar".

Line 141-142: Change "raising the possibility that they evolved" to "possibly evolving".

Line 142: Change "harbour" to "harbor".

Line 154: Change "hypervirulence determining" to "hypervirulence-determining".

Line 155: Change "are" to "is".

Line 161: Change "hospital acquired" to "hospital-acquired".

Line 162: Change "South-East" to "Southeast".

Line 164: In "Despite of this, hvKP infection was successfully managed...", remove "of".

Line 167: Change "of" to "for".

Line 168, 171, 173: Change "carbapenem resistant" to "carbapenem-resistant".

Line 169: In "and from other", delete "from".

Line 173: Change "higher" to "a higher".

Line 179: correct "centre".

Line 182: Delete "in".

Sometimes, "hvKP" is used interchangeably with "K. pneumoniae," which could confuse. Consider ensuring consistency in terminology.

Although the study focuses on molecular epidemiology, it is necessary to discuss the severity of the infections caused by these three isolates, which would highlight their clinical relevance.

It would also be valuable to review whether these clones have been associated with severe infections or outbreaks in other European countries.

Regarding the Pécs isolate KP48326 ST86, which resulted from the same cluster as a Greek isolate (ID-48733), consider performing a genetic distance analysis to determine its relationship.

The conclusions highlight the importance of molecular surveillance; specific surveillance strategies for Hungary could be suggested.

MATERIAL Y METODOS

Line 186: Were only the first isolates included in the case of recurrent infections, or were duplicate samples from the same patient excluded?

Line 187: Substitute "the University" to "of the University".

Line 192: Remove "for".

Line 193: Did the string test yield any unclear findings for any of the 157 strains?

Line 195: Replace "were" to "was".

Line 197: "ampicillin" is correct.

Line 200: While EUCAST suggest the broth microdilution method, the use of disk diffusion is suggested for cefiderocol. The authors should update the technique to comply with EUCAST guidelines or provide justification for the decision."

Line 204: It would be useful to include the average coverage obtained of the genomes and the quality standards for approving whole-genome sequencing analyses.

For the comparison with international isolates, was a specific criterion used to define close relationships in the cgMLST analysis?

Indicate whether additional manual validation of relevant plasmids was performed (e.g., BLASTn against specialized databases).

RESPONSE TO REVIEWERS Spectrum00031-25 (Hypervirulent *Klebsiella pneumoniae* causing bloodstream infections in Hungary)

We are grateful for the efforts of the reviewers to improve our manuscript. Please, see our responses and comments below. Please note that the number of lines indicating our changes, additions *etc.* are referring to those of the updated version of the manuscript.

Reviewer #1

Major Concerns

1. I would encourage the authors to include their takeaway from their findings and what the implications are. hvKP numbers were low in Hungary compared to China, South Korea, Taiwan, and South Africa - any ideas as to why? Same comment but for the similarity to Italy. What should epidemiologists do as a result of these findings? Should we be screening for hvKP genes? Since these infections are generally community acquired, what approach should be taken to study them?

Thank you for these suggestions and questions. They have been addressed in the revised version of the manuscript in lines:

152-160

The epidemiological difference could partly be explained by increased colonisation and transmission, and possible, but not proven genetic predisposition of people living in or originating from Southeast Asian countries (34).

The hvKP isolates identified in this study belonged to established hypervirulent clones, i.e. ST5, ST86 and ST6771, the latter one being a single locus variant of ST893. Notably, we did not identify any hvKP ST23, i.e. the most common hvKP clone in Southeast Asia, and also found in other European countries (Germany, Italy, Poland, Switzerland, France, Ireland, Lithuania, Latvia and the Netherlands) (27, 35).

220-223

*Such surveillance based on testing of all *K. pneumoniae* from bloodstream infections, and other likely infectious sites, e.g. liver abscess, endophthalmitis, cerebrospinal fluid using genetic markers would be beneficial to assess the true burden of hvKP in Hungary and other countries with low prevalence, so far.*

Minor Concerns

1. A comment about limiting study isolates to BSI would be beneficial for context.

A sentence was added in lines 160-162

However, as our collection was restricted to bloodstream isolates, the possibility of hvKP ST23 being present in liver abscess, cerebrospinal fluid, necrotizing fasciitis or endophthalmitis samples cannot be excluded.

2. Was the hospital environment near these patients sampled to determine if the hvKP had colonized any surfaces or wastewater drains?

Unfortunately, we are not aware of any environmental sampling near the patients. As the data presented were generated retrospectively, the infections were mostly community acquired, and the bloodstream isolates were susceptible, the IPC team was not alerted.

Reviewer #2

ABSTRACT

Line 20: "There is no data" change to "no data are available". - *Corrected as requested.*

Line 22: Add a comma after "Hungary" - *Corrected as requested.*

Lines 23-24: "three hvKP isolate was determined" correct to "three hvKP isolates were sequenced". - *Corrected as requested.*

Line 24: Add a comma after "ST86". - *Corrected as requested.*

Line 26: Add a comma after "isolates". - *Corrected as requested.*

Line 32 "although epidemiological link could not be established..." modify to "although no direct epidemiological link could be established..." - *Modified as requested.* Line 35: "

Highlights the importance of continued whole genome based epidemiological surveillance" modify to " Underscores the importance of continued whole-genome-based epidemiological surveillance". - *Modified as requested.*

IMPORTANCE

Line 43: "The localisation of hypervirulence related genes" correct to "The localization of hypervirulence-related genes". - *Corrected as requested.*

Line 42: Correct "hvKp" to "hvKP". -*Corrected as requested.*

Line 43-44: Rewrite the following text: "Highly similar to virulence plasmids or chromosomally integrated ICEKp1 of *K. pneumoniae* ST23 and ST86 isolated in other continents". - *Modified as requested.*

Line 46 I suggest changing "emphasizes the additional role of horizontal gene transfer" to "Underscores the significant role of horizontal gene transfer in the spread of hvKP." - *Modified as requested.*

INTRODUCTION

Line 50: "a Gram-negative bacterium causes" change to "is a Gram-negative bacterium that causes....". - *Modified as requested.*

Line 51: Add a comma after "urinary tract and infections". - *Corrected as requested.*

Line 52: "the virulence of the actual *K. pneumoniae* strain" modify to "the virulence of a given *K. pneumoniae* strain". - *Modified as requested.*

Line 60 "HvKP typically causes community acquired infections" change to "HvKP typically causes community-acquired infections". - *Corrected as requested.*

Line 61: Add a comma after "endophthalmitis". - *Corrected as requested.*

Lines 80-81: "However, recently the emergence of multi-drug resistant hvKP isolates has been reported mostly from China" modify to "However, in recent years, multidrug-resistant hvKP isolates have emerged, particularly in China". - *Modified as requested.*

Line 66: "of capsular polysaccharide" correct to " of capsular polysaccharides". - *Corrected as requested.*

Although the study focuses on molecular epidemiology, I suggest, it is necessary to expand on the clinical relevance of hvKP by providing information on prevalence rates, mortality, and morbidity of hvKP infections, as well as their therapeutic challenges, particularly in MDR strains.

This point has been addressed in lines:

61-65

The mortality of hvKP bacteraemia is significantly higher compared to bacteraemia caused by non-hvKP (6). Cases of community-acquired pneumonia with bacteremia and necrotizing fasciitis have also presented with high mortality rates of 55 and 47%, respectively. Furthermore, HvKP infections often cause permanent loss of vision or neurologic sequelae (7).

72-78

The true prevalence of hvKP is difficult to assess as studies define hvKP based on various combinations of markers. Nevertheless, when considering only genetic marker defined hvKP prevalence among bloodstream isolates, it was highest in China, where it reached 73.9% in certain regions (18) and in South and Southeast Asian countries (around 20%) (19), whereas in North-America and Europe it was found to be invariably <10 % (3.8% in Chicago, USA (20), 8.2% in Canada (21) 3.2% in Barcelona, Spain (22), 1.3% in Italy (23), and 3.3% in Croatia (24)).

81-85

The convergence of resistance and virulence poses a formidable challenge to treatment, especially when a hvKP is producing a carbapenemase enzyme, and such carbapenem resistant hvKP belonging to sequence type ST23 also occurred in several European countries, including one case being reported in 2023 from Hungary, as well (27).

Information on the prevalence of *K. pneumoniae* infections in Hungarian hospitals should be added, along with any prior reports of hvKP in the region. These data and their comparison with other regions would provide stronger justification for the study in

Hungary.

We tried to recover nation-wide information on the prevalence of *Klebsiella pneumoniae* in bloodstream isolates, but such information is not available. Nevertheless, the data on the prevalence of *K. pneumoniae* in bloodstream infections in our institution during the study period is given in lines

87-90

Therefore, the aim of our investigation was to assess the presence and characteristics of hvKP, to determine its proportion among bloodstream isolates in our university hospital, where Klebsiella pneumoniae was a significant pathogen being isolated from 8.7% of true positive blood cultures in the study period.

The regional situation is partly discussed in the answer to the previous point, and also discussed in lines

146-152

This figure is comparable to multi-center data from Italy (1.3%) (23), but lower than what was observed in Croatia (3.3%) (24), a country bordering Southwestern Hungary, i.e. the region our university hospital is located, and markedly lower than what was reported from China, South Korea, Taiwan and South Africa (22.8 %, 42.2%, 46% and 52%, respectively) (31-33). It is of note though that, to the best of our knowledge, other countries in Central Europe, e.g. Austria, Slovakia, Romania, Serbia have not reported the presence of hvKP, yet.

RESULTS

Line 80: Clarify that the hypermucoviscous phenotype alone is not sufficient to define hvKP, as there are hypermucoviscous strains that are not hypervirulent and vice versa.

The paragraph was re-phrased to clarify this point in lines 94-98

"...nevertheless twelve of them did not possess the rmpA/A2 and iucA genes, which were present in two string test positive isolates (KP48326 and KP873), and in another non-hypermucoviscous isolate (KP7389). As the hyperviscous character alone does not define hvKP (11), only the three rmpA/A2 and iucA PCR positive strains were selected for further studies."

Line 99: Delete "as well as" and replace it with "were recovered and". - *Modified as requested.*

Line 100: Correct "Symptoms of infections from which the hvKP strains was isolated" to "Infections caused by hvKP strains" - *Corrected as requested.*

Line 102: Change community acquired" to "community-acquired". - *Corrected as requested.*

Lines 102-103: To improve readability, I suggest changing "Signs of a bloodstream infection of the third patient emerged and the sample was collected on the fourth day after admission" to "Bloodstream infection symptoms in the third patient appeared on the fourth day after admission". - *Modified as requested.*

Line 103: Change "hospital acquired" to "hospital-acquired". - *Corrected as requested.*

Lines 109-110: For clarity, I suggest changing: "Apart from being resistant to ampicillin, all three hvKP isolates remained susceptible to all antibiotics tested and none of them carried any acquired antibiotic resistance genes" to "Except for ampicillin, all three hvKP isolates were susceptible to all tested antibiotics and did not carry any acquired antimicrobial resistance genes". - *Modified as requested.*

Line 113: Change "hypervirulence associated" to "hypervirulence-associated". - *Corrected as requested.*

Line 119: Change "Comparison" to "A comparison". - *Corrected as requested.*

Since the three hvKP strains were phenotypically resistant only to ampicillin, no convergence of hypervirulence with multidrug resistance was observed. It is necessary to provide patient data to determine whether they developed sepsis or septic shock, if they had complications such as metastatic abscesses, meningitis, or endophthalmitis, and their clinical outcomes.

This point is clarified now in Table 1. and in lines 105-107:

Although all three patients had sepsis, none developed septic shock or metastatic infections, and the sepsis was manageable with the empiric antibiotic therapy shown in Table 1.

DISCUSSION

Line 143-144: Change "have been increasing" to "have increased". - *Corrected as requested.*

Line 144: Change "hvKP identified" to "hvKP is identified".

Respectfully disagree, as the statement is about the prevalence of hvKP being 2%. In order to improve clarity, the sentence was re-phrased to:

*Here, we report that in our clinical settings in Hungary the prevalence of hvKP (defined by the possession of capsular overproduction (*rmpA/A2*) and aerobactin (*iutA-iucABCD*) genes) is 2% among *K.pneumoniae* bloodstream isolates.*

Line 171: Change "similarly" to "similar". - *Corrected as requested.*

Line 175: Change "raising the possibility that they evolved" to "possibly evolving". - *Modified as requested.*

Line 176: Change "harbour" to "harbor". - *Corrected as requested.*

Line 187: Change "hypervirulence determining" to "hypervirulence-determining". - *Corrected as requested.*

Line 188: Change "are" to "is". *Corrected as requested.*

Line 196: Change "hospital acquired" to "hospital-acquired". - *Corrected as requested.*

Line 197: Change S"outh-East" to "Southeast". - *Corrected as requested.*

Line 199: In "Despite of this, hvKP infection was successfully managed...", remove "of". - *Corrected as requested.*

Line 168, 171, 173: Change "carbapenem resistant" to "carbapenem-resistant". *Corrected throughout the text.*

Line 206: In "and from other", delete "from". - *This sentence has been re-phrased.*

Line 211: Change "higher" to "a higher". - *Corrected as requested.*

Line 206: correct "centre". - *"centre" has been corrected to "center" throughout the text except in the name of the ECDC (<https://www.ecdc.europa.eu/en>), since this is the official name of the European Centre for Disease Prevention and Control.*

Line 207: Delete "in". - *This sentence has been re-phrased.*

Sometimes, "hvKP" is used interchangeably with "K. pneumoniae," which could confuse. Consider ensuring consistency in terminology.

The text was revised, and now hvKP is used where it is appropriate.

Although the study focuses on molecular epidemiology, it is necessary to discuss the severity of the infections caused by these three isolates, which would highlight their clinical relevance.

The information requested is now included in Table 1, in lines 105-107 (see above) and the discussion too, in lines 199-203:

Despite this, hvKP pulmonary infections and/or sepsis was successfully managed with the empirical antibiotic therapy, and septic shock or metastatic infections did not occur in them. The single patient who succumbed did so due to severe SARS-CoV-2 pulmonary superinfection likely acquired during hospitalization, as she tested negative for SARS-CoV-2 at admission and became positive only on the 25th day of her hospital stay.

It would also be valuable to review whether these clones have been associated with severe infections or outbreaks in other European countries.

The information requested is now included in lines 162-170.

Of the hvKP clones identified in this study, ST86 is the most widely recognized one reported among others from Croatia (24), France (36), Spain (22), Japan (37), China (38) and the USA (39). Although cgMLST and genomic analysis identified a Greek hvKP ST86 closely related to KP48326, our patient's history did not reveal any connection to Greece. Publication on HvKP ST86 in Europe is available on sporadic cases detected in France, Spain, Italy and Croatia (22-24, 40). HvKP ST893 is a predominant clone in Iran, and although the Pasteur BigSdb-Database contains hvKP genomes from Belgium and Norway (Suppl. Table 1), to the best of our knowledge, no publication describes such isolates in Europe (28).

And also, in lines 188-190.

Hypervirulent K. pneumoniae of this sequence type is seldom reported, mainly from Italy (23, 41), but in the Pasteur BigSdb database few genomes of hvKP isolates from the United Kingdom, Ireland, Spain and Slovenia are deposited (Suppl. Table 1.).

Regarding the Pécs isolate KP48326 ST86, which resulted from the same cluster as a Greek isolate (ID-48733), consider performing a genetic distance analysis to determine its relationship.

Thank you for the suggestion. We performed the analysis, and this information is included in the current version of the manuscript in lines 123-127.

"Further assessing the genomic relatedness of KP48326 and id-48733 by Mash (d-distance), FastANI, and core-genome single-nucleotide polymorphism (SNP) analysis consistently indicated a high degree of genomic similarity, with a low d-distance (0.000336872), high FastANI identity of 99.9977%, and only 43 SNP differences, collectively supporting that the genomes are indeed highly similar."

The methodology is described in lines 258-262.

Genetic distance of isolates from this study and from the Bigsdb database clustering with ≤ 15 allelic differences was analyzed using three complementary approaches: Mash (v2.1) for estimating genetic distance (d-distance)(48), FastANI (v1.34) for pairwise nucleotide similarity (<https://github.com/ParBLiSS/FastANI/releases/tag/v1.34>), and Snippy (v4.0.2) for single-nucleotide polymorphism (SNP) (<https://github.com/tseemann/snippy>).

The conclusions highlight the importance of molecular surveillance; specific surveillance strategies for Hungary could be suggested.

Thank you for the comment , this suggestion is now included in lines 220-223:

Such surveillance based on testing of all K. pneumoniae from bloodstream infections, and other likely infectious sites, e.g. liver abscess, endophthalmitis, cerebrospinal fluid using genetic markers would be beneficial to assess the true burden of hvKP in Hungary and other countries with so far low prevalence.

MATERIALS AND METHODS

Line 186: Were only the first isolates included in the case of recurrent infections, or were duplicate samples from the same patient excluded?

Only the first isolate from a patient was included into the study. This information is provided now in line 227.

Line 228: Substitute "the University" to "of the University". - *Corrected as requested.*

Line 234: Remove "for". - *Corrected as requested.*

Line 235: Did the string test yield any unclear findings for any of the 157 strains?

The string test could be evaluated as positive or negative for all isolates using the ≥ 5 mm string threshold for positivity. However, the majority of string test positive isolates did not have the hypervirulence defining genes, as detailed in the response to the first question in the results section.

Line 237: Replace "were" to "was". - *Corrected as requested.*

Line 239: "ampicillin" is correct. - *Corrected as requested.*

Line 243: While EUCAST suggest the broth microdilution method, the use of disk diffusion is suggested for cefiderocol. The authors should update the technique to comply with EUCAST guidelines or provide justification for the decision."

We respectfully disagree with the above statement.

The EUCAST does not suggest broth microdilution method being superior to disc diffusion for those antibiotics we tested. In the study we followed the EUCAST prescribed method with the media, discs and control strains described in EUCAST documents

(https://www.eucast.org/fileadmin/src/media/PDFs/EUCAST_files/Disk_test_documents/2025_manuals/Manual_v_13.0_EUCAST_Disk_Test_2025.pdf)

(https://www.eucast.org/ast_of_bacteria/quality_control), and interpreted the diameters based on the clinical breakpoints v.14 published for Enterobacterales (

https://www.eucast.org/fileadmin/src/media/PDFs/EUCAST_files/Breakpoint_tables/v14.0_Breakpoint_Tables.pdf valid at the time of study) for all antibiotics tested.

However, unfortunately tetracycline was also listed by mistake as an antibiotic tested, which was incorrect, as it cannot be interpreted either by disc diffusion or by MIC measurement based on the EUCAST. So, the tetracycline was removed from the sentence, which reads now in lines 239-244:

The susceptibility to ampicillin, ampicillin-sulbactam, amoxicillin-clavulanate, piperacillin-tazobactam, cefuroxime, cefotaxime, ceftazidime, ceftriaxone, cefepime, ceftazidime-avibactam and ceftolozane-tazobactam, ertapenem, imipenem, meropenem, ciprofloxacin, levofloxacin, gentamicin, tobramycin, amikacin, trimethoprim-sulfamethoxazole, and cefiderocol of strains identified as hvKP was tested by Kirby-Bauer disk diffusion method on Mueller-Hinton agar (BioRad, Marnes-la-Coquette, France) according to the EUCAST guidelines (36).

Line 249: It would be useful to include the average coverage obtained of the genomes and the quality standards for approving whole-genome sequencing analyses.

We are thankful for the suggestion. This information is now provided in Supplementary Table 2.-7. As indicated in lines 250-250 in the revised manuscript.

For the comparison with international isolates, was a specific criterion used to define close relationships in the cgMLST analysis?

We used the cluster distance threshold of 15 or less alleles. This information is now provided in the legend of Figure 2., and in the text in lines 258-259:

Genetic distance of isolates from this study and from the Bigsdb database clustering with ≤ 15 allelic differences was analyzed using three complementary approaches

Indicate whether additional manual validation of relevant plasmids was performed (e.g., BLASTn against specialized databases).

In the revised version it is clarified in lines 272-274.

Subsequently, we performed nucleotide BLAST comparisons involving our plasmids, and those belonging to the same PTU in the Proksee pipeline. All plasmid sequences were obtained from the NCBI database as shown in Table 3.

Re: Spectrum00031-25R1 (Hypervirulent *Klebsiella pneumoniae* causing bloodstream infections in Hungary)

Dear Dr. Agnes Maria Sonnevend:

Your manuscript has been accepted, and I am forwarding it to the ASM production staff for publication. Your paper will first be checked to make sure all elements meet the technical requirements. ASM staff will contact you if anything needs to be revised before copyediting and production can begin. Otherwise, you will be notified when your proofs are ready to be viewed.

Sincerely,
Ana Cabrera
Editor
Microbiology Spectrum

Reviewer #1 (Comments for the Author):

N/A